# CO₂-Assisted Dehydrogenation of Propane to Propene over Zn-BEA Zeolites: Impact of Acid–Base Characteristics on Catalytic Performance

Svitlana Orlyk [1], Pavlo Kyriienko [1], Andriy Kapran [1], Valeriy Chedryk [1], Dmytro Balakin [2], Jacek Gurgul [3], Malgorzata Zimowska [3], Yannick Millot [4] and Stanislaw Dzwigaj [4,*]

[1] L.V. Pisarzhevskii Institute of Physical Chemistry, National Academy of Sciences of Ukraine, 31 Prosp. Nauky, 03028 Kyiv, Ukraine
[2] Institute of Physics, National Academy of Sciences of Ukraine, 46 Prosp. Nauky, 03028 Kyiv, Ukraine
[3] Jerzy Haber Institute of Catalysis and Surface Chemistry, Polish Academy of Sciences, Niezapominajek 8, PL-30239 Kraków, Poland
[4] Laboratoire de Réactivité de Surface, Sorbonne Université-CNRS, UMR 7197, 4 Place Jussieu, F-75005 Paris, France
[*] Correspondence: stanislaw.dzwigaj@sorbonne-universite.fr

**Abstract:** Research results about the influence of BEA zeolite preliminary dealumination on the acid–base characteristics and catalytic performance of 1% Zn-BEA compositions in propane dehydrogenation with CO₂ are presented. The catalyst samples, prepared through a two-step post-synthesis procedure involving partial or complete dealumination of the BEA specimen followed by the introduction of Zn²⁺ cations into the T-positions of the zeolite framework, were characterized using XRD, XPS, MAS NMR, SEM/EDS, low-temperature N₂ ad/desorption, C₃H₈/C₃H₆ (CO₂, NH₃)-TPD, TPO-O₂, and FTIR-Py techniques. Full dealumination resulted in the development of a mesoporous structure and specific surface area (BET) with a twofold decrease in the total acidity and basicity of Zn-BEA, and the formation of Lewis acid sites and basic sites of predominantly medium strength, as well as the removal of Brønsted acid sites from the surface. In the presence of the ZnSiBEA catalyst, which had the lowest total acidity and basicity, the obtained selectivity of 86–94% and yield of 30–33% for propene (at 923 K) exceeded the values for ZnAlSiBEA and ZnAlBEA. The results of propane dehydrogenation with/without carbon dioxide showed the advantages of producing the target olefin in the presence of CO₂ using Zn-BEA catalysts.

**Keywords:** Zn-BEA zeolites; dealumination; acid–base characteristics; propane dehydrogenation with CO₂; propene

## 1. Introduction

The world production of propene—the raw material for the synthesis of polypropylene and many important organic compounds (propene oxide, acrylic acid, propylene glycol, etc.)—exceeds 100 million tons per year. Conventional propene production through the steam cracking or catalytic cracking of petroleum does not meet the growing market needs. The catalytic dehydrogenation of propane, especially direct (PDH) and oxidative dehydrogenation using O₂ or N₂O, and CO₂-mediated dehydrogenation (CO₂-PDH) are considered a promising alternative to the oil-based cracking process [1,2]. The participation of CO₂ in the dehydrogenation of alkanes is of interest as a potential approach to utilizing carbon dioxide [3–5]. An important task for the realization of these processes is the development of active and selective catalysts that are not rapidly deactivated (especially in direct dehydrogenation) by coking.

Besides metal oxide catalysts for PDH or CO₂-PDH, zeolites containing cations or oxide nanoparticles of active components (mainly Cr, Ga, and Pt-Sn as components of known metal oxide catalysts) are of great interest for the dehydrogenation of alkanes [1,4,6–12].

It is known that the catalytic properties of zeolite catalysts are largely determined by their acid–base characteristics, which, in turn, depend on the Si/Al ratio, the nature of the active component, its form (nanoparticles, clusters, isolated cations), and quantity/density. In particular, partial or complete dealumination of the zeolite has been shown to increase the catalyst selectivity for the target product, especially for the dehydrogenation of propane in the presence of $CO_2$. This is due to changes in the acid–base properties of the system, in particular, a reduction in the concentration of Brønsted acid sites (BAS) until they are removed from the surface—as a result of complete dealumination—and changes in the form of the active component, in particular, the formation of isolated cations or the stabilization of oxidized subnanoclusters of the active component in/near vacant positions in the dealuminated zeolite [7,13–16].

A number of recent works have shown the potential of eco-benign Zn-containing catalysts for the dehydrogenation of alkanes, in which ZnO nanoparticles applied to the zeolite act as the main component [13–15,17–19] or zinc species as isolated cations and ZnO clusters are cocatalysts in bimetallic systems (Pt-Zn, Cr-Zn, Ni-Zn) based on high-silica zeolites [20–24]. Thus, studies of Zn-containing zeolites have used samples containing zinc oxide in amounts of ZnO $\geq$ 3–20 wt % [13–15,18]; in research on bimetallic systems with isolated Zn(II) forming Lewis acid sites (LAS), the main focus is on the effect of zinc as a transition metal cocatalyst. At the same time, the promoting effect of $Zn^{2+}$ cations in $Zn^{2+}$/H-BEA and the synergistic effect of Zn sites and BAS on the activation of the C−H bonds of methane is stronger than that of ZnO species in ZnO/H-BEA [25]. Therefore, it can be expected that Zn-containing zeolites with isolated Zn (II) may also be of interest as catalysts for propane dehydrogenation.

Despite the progress in the study of Zn-containing zeolite catalysts for DH or $CO_2$-DH processes of lower alkanes, it is not yet known how the location of zinc as isolated atoms will affect their catalytic properties. The effect of the Si/Al ratio on the catalytic properties of Zn-containing zeolites with BEA structure, which, according to Zhao et al. [18], dominate over catalysts based on zeolites with other structural types in terms of activity and selectivity to propene, has also not been clarified.

The $CO_2$-PDH process on the catalysts, which do not undergo redox transformations under reaction conditions (including Ga-, Zn-containing), is considered to be mainly direct propane dehydrogenation (1), the equilibrium of which shifts in the direction of propene production due to the consumption of hydrogen in the reverse water–gas shift reaction (RWGSR) (2) [1–4]:

$$C_3H_8 \leftrightarrow C_3H_6 + H_2 \qquad \Delta H_{298K} = + 124 \text{ kJ mol}^{-1} \qquad (1)$$

$$CO_2 + H_2 \leftrightarrow CO + H_2O \qquad \Delta H_{298K} = + 41 \text{ kJ mol}^{-1} \qquad (2)$$

Carbon dioxide may also participate in coke gasification through the reverse Boudouard reaction $CO_2 + C \leftrightarrow 2CO$ ($\Delta H_{298K} = \Delta H_{298K} = + 172 \text{ k Jmol}^{-1}$) that enhances the catalyst stability.

In this paper, we report on the influence of the preliminary dealumination of BEA zeolite on the acid–base characteristics of synthesized Zn-BEA samples and their catalytic properties in the $CO_2$-mediated dehydrogenation of propane to propene. Zn-BEA specimens with different Si/Al ratios and a zinc loading of 1 wt % were prepared using a two-step post-synthesis procedure including preliminary partial and full dealumination of initial BEA zeolite followed by the incorporation of zinc cations into the vacant T-atom sites of the zeolite framework. The catalytic behavior of the Zn-BEA zeolites in propane dehydrogenation was tested both in the presence and absence of carbon dioxide in the initial reaction mixture.

## 2. Results and Discussion

### 2.1. Structure, Texture, and Acid–Base Characteristics of Zn-BEA Zeolites with Different Si/Al Ratios

The presence of diffraction peaks typical of BEA zeolites in the corresponding XRD patterns indicates that the dealumination of the initial TEABEA sample (Si/Al = 17) with nitric acid and subsequent incorporation of zinc cations into the SiBEA framework does not affect the crystallinity of the structure, as shown in Figure 1. The increase in the unit cell parameter $d_{302}$ to 3.976 Å (ZnSiBEA; $2\theta$ = 22.36°) compared to 3.920 Å (SiBEA; $2\theta$ = 22.68°) is due to the expansion of the BEA zeolite matrix as a result of the interaction of zinc ions with OH groups of vacant T-atom sites and, as a consequence, their incorporation into the zeolite structure, resulting in an increase in the Zn-O bond length compared to Si-O or Al-O.

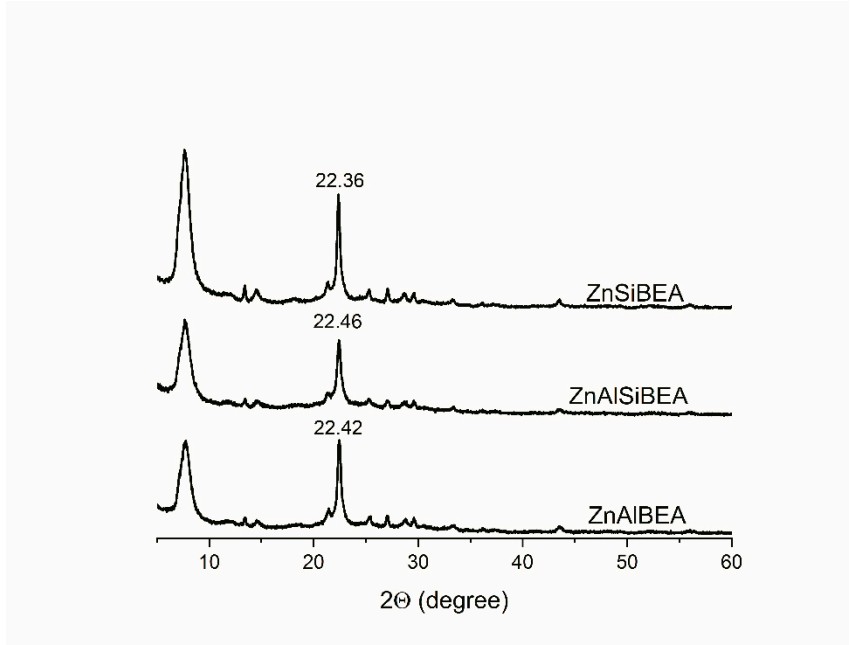

**Figure 1.** X-ray diffraction patterns of Zn-BEA samples after calcination at 923 K.

Each studied Zn-BEA sample is characterized by the presence of micropores with a total volume of ~0.2 cm$^3$/g and an average diameter of ~1 nm, as given in Table 1. The volume of mesopores of the samples is 0.32–0.37 cm$^3$/g. As can be seen from the data presented, the complete dealumination led to the formation of mesopores with the largest diameter/surface of ~60 nm/80 m$^2$/g and specific surface area (BET) of 605 m$^2$/g.

**Table 1.** Texture characteristics of zeolite Zn-BEA samples.

| Sample | Micropores | | Mesopores | | | $S_{BET}$, m$^2$/g | Adsorption Volume at $p/p_0$ = 1, cm$^3$/g |
| | Volume $V_{mi}$, cm$^3$/g | Diameter $d_{mi}$ *, nm | Volume $V_{me}$, cm$^3$/g | Diameter $d_{me}$, nm | $(S_{me} + S_{outer})$, m$^2$/g | | |
|---|---|---|---|---|---|---|---|
| ZnAlBEA | 0.19 | 1.00 | 0.37 | 32 ± 5 | 70 | 535 | 0.58 |
| ZnAlSiBEA | 0.18 | 1.01 | 0.32 | 50 ± 15 | 60 | 505 | 0.52 |
| ZnSiBEA | 0.21 | 1.05 | 0.33 | ~60 * | 80 | 605 | 0.56 |

* Determined through the Saito–Foley method.

In Figure S1, SEM images with a magnification of 25,000 (left) and 50,000 (right) of ZnSiBEA (Figure S1a,b,) ZnAlSiBEA, (Figure S1c,d) and ZnAlBEA (Figure S1e,f) are presented. They illustrate the morphology of the examined samples. SEM/EDS analysis

reveals that they are composed of Si, O, Zn, and Al, and the amount of Al increases from 0.4 wt % in ZnSiBEA to 1.2 wt % in ZnAlSiBEA and 2.6 wt % in ZnAlBEA. The amount of Zn is close to 1.2 wt %; however, ZnSiBEA exhibits the highest quantity of zinc (1.4 wt %). It is worth noting that the ZnSiBEA sample exhibits the presence of the largest 160–100 nm particles, while ZnAlSiBEA led to crystallites not exceeding 120 nm. The morphology of the ZnAlBEA particles is very similar and in the range of 150–100 nm.

The relative abundance of elements on the surface of the Zn-BEA catalysts obtained from the XPS survey scans in the depth of max. 11.2 nm are presented in Table 2.

**Table 2.** Surface elemental composition of Zn-BEA zeolites (at. %).

| Sample | Zn | Si | Al | O | C |
|---|---|---|---|---|---|
| ZnAlBEA | 0.23 | 33.24 | 2.07 | 58.53 | 5.92 |
| ZnAlSiBEA | 0.25 | 34.53 | 0.59 | 59.68 | 4.95 |
| ZnSiBEA | 0.24 | 36.86 | 0.11 | 57.95 | 4.84 |

These were computed with the assumption that the samples are made of pure and uniform $SiO_2$ with a density equal to 2.18 g cm$^{-3}$ [26]. The Si/Al ratios calculated for ZnAlBEA (16), ZnAlSiBEA (58), and ZnSiBEA (335) prove strong dealumination of BEA zeolite through the two-step post-synthesis procedure. High-resolution spectra of Zn 2p, Si 2p, Al 2p, O 1s, and C 1s were used to investigate the chemical states of the active phase in the catalysts.

The C 1s core lines (Figure 2A) of the Zn-BEA catalysts are composed of three characteristic peaks at 285.0 eV (organic contaminants), 285.9–286.3 eV (C-O groups), and 289.9–290.5 eV (O–C=O groups). ZnAlBEA has an additional fourth component with a BE of 288.1 eV (13%) related to C=O groups. The dealuminated catalyst (ZnSiBEA) shows significantly lower content of C–O groups (28%) compared to the others (42–44%), whereas the amount of O–C=O groups does not exceed 8% in any sample (Table S1 in Supplementary Materials). The hydrocarbon contamination was used as an internal calibration for XPS spectra, as mentioned below in Section 3.1.

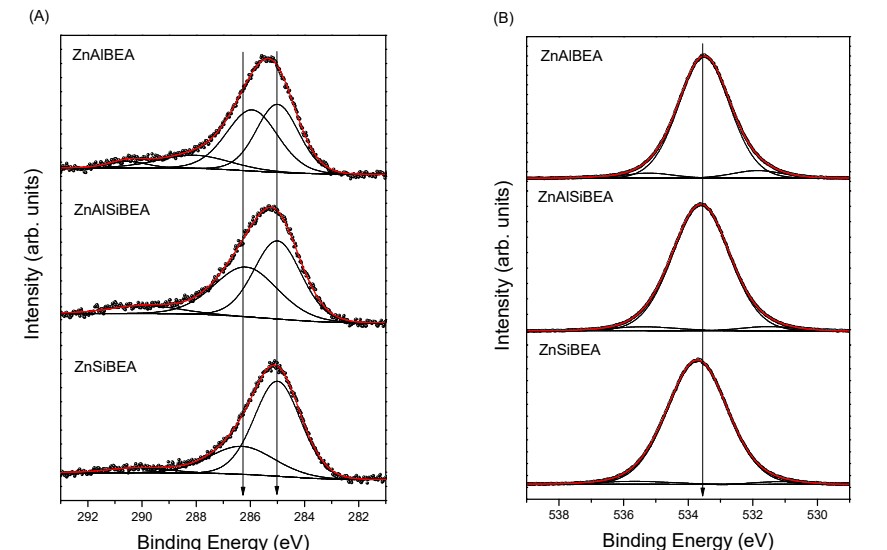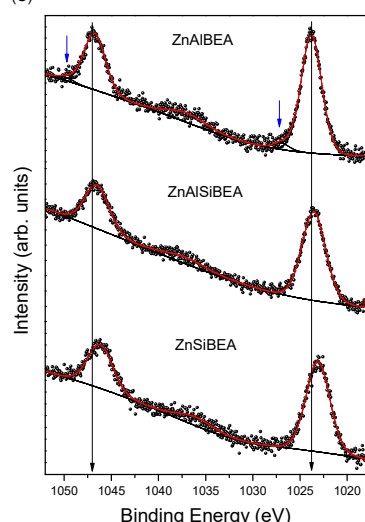

**Figure 2.** XPS spectra of Zn-BEA zeolites in the BE regions of C 1s (**A**), O 1s (**B**), and Zn 2p (**C**). Lines marking the most intense components. Blue arrows show an additional component in the Zn 2p spectrum of ZnAlBEA.

The O 1s spectra (Figure 2B) of the catalysts show three components: (i) a main-line (over 90% of total spectrum area) at BE of 533.5–533.7 eV related to oxygen located in the BEA zeolite lattice [27–30], (ii) oxygen from defective sites of the zeolite matrix

(BE = 531.2–531.8 eV) [31–33], and (iii) a peak at BE > 535 eV assigned to physisorbed water, and/or the oxygen of organic contaminants (Table S1 in Supplementary Materials). The relative decrease in aluminum in the catalysts causes an increase in lattice components at the cost of reducing the area of the low-BE component. This strongly suggests that as well as Si-O-Zn bonds, Al-O bonds also contribute to this component. However, a slightly lower BE range of the O 1s line (530.5–531.3 eV) was attributed to the thin films of $Al_2O_3$ [34,35]. The peak corresponding to the Zn–O bond (529.9 eV) [31,36] was not observed.

A single silicon component was detected with a Si $2p_{3/2}$ BE value close to 104.0 eV in ZnAlBEA, 104.2 eV in the partially dealuminated catalyst, and 104.3 eV in ZnSiBEA (Figure S2), and a single aluminum component was detected with an Al $2p_{3/2}$ BE value close to 75.6 eV in ZnAlBEA, 75.5 eV in the partially dealuminated catalyst, and 75.0 eV in ZnSiBEA (Table 3) (Figure S3). Such a contribution can be associated with the $Al^{3+}$ in tetrahedral positions similar to the case of Faujasites [37]. The spin-orbit splitting of the Al 2p doublet was constrained to $\Delta_{SO} = 0.41$ eV.

**Table 3.** XPS data of Zn-BEA zeolites.

| Core Excitation | ZnAlBEA | | ZnAlSiBEA | | ZnSiBEA | | |
|---|---|---|---|---|---|---|---|
| | BE (eV) | Area (%) | BE (eV) | Area (%) | BE (eV) | Area (%) | |
| Zn 2p3/2 | 1023.8 1027.0 | 95.9 4.1 | 1023.5 | 100 | 1023.1 | 100 | A B |
| Si 2p3/2 | 104.0 | 100 | 104.2 2 | 100 | 104.3 | 100 | |
| Al 2p3/2 | 75.6 | 100 | 75.55 | 100 | 75.0 | 100 | |

Figure 2C presents Zn 2p core-level spectra obtained for the ZnAlBEA, ZnAlSiBEA, and ZnSiBEA samples. They can be well fitted by a single symmetric doublet with fairly high Zn $2p_{3/2}$ BE values of 1023.1–1023.8 eV. It should be noted that the Zn $2p_{3/2}$ line for ZnO was quoted at 1022.0 eV [36], whereas metallic Zn was reported at a BE of 1021.6 eV [38]. It is clear that the BE shift observed between bulk ZnO and our Zn-BEA zeolites indicates different electronic states of Zn. Such an increase in BE could be a result of the incorporation of Zn into the zeolite lattice. In this case, Zn species are localized at the vacant T-atom sites, where the oxygen from the zeolite matrix exhibits higher electronegativity than the $O^{2-}$ ligand in bulk ZnO. This results in a reduction in the valence electron density of Zn in the Zn–O–Si bond and an increased binding energy [31,39]. One can note that a similar effect was observed in several zeolites doped with Zn, e.g., MFI [31], ZSM-5 [25,40], FAU [37], BEA [41,42], and Y zeolites [43]. All these papers attribute the Zn $2p_{3/2}$ peak at about 1023.2 eV to the isolated $[Zn(OH)]^+$ species, which are formed from the tight interactions of zinc species with BEA zeolite. Such $[Zn(OH)]^+$ species can decompose to form water and $[Zn–O–Zn]^{2+}$ [25,42,44]. Therefore, the Zn 2p doublets can be reasonably assigned to the Zn(II) species located in the framework of the BEA zeolite with tetrahedral symmetry. This is also confirmed by the spin-orbit splitting of 23.0 eV characteristic of divalent Zn species.

Moreover, one can identify a very small component (4%) in ZnAlBEA with a BE of 1027.0 eV related to unknown Zn(II) species. However, it has been stated that wet chemistry-based techniques can lead to the incorporation of various Zn species into the zeolite, including isolated $Zn^{2+}$ or $[Zn(OH)]^+$ cations localized at the exchange positions, as well as binuclear $[Zn–O–Zn]^{2+}$ or multinuclear $[Zn–(O–Zn)_n]^{2+}$ clusters [39,44,45]. Perhaps cluster formation is the reason for the appearance of this additional component.

All Si 2p spectra are well fitted by a single doublet with a spin-orbit splitting of 0.61 eV. The high binding energies of Si $2p_{3/2}$ shown in Table 3 prove that only Si(IV) species are present in our BEA catalysts. It is worth mentioning that these values are slightly larger than those reported for MFI and MOR zeolites elsewhere. The dealumination process does not cause the appearance of an additional component in the Si 2p spectra, but only a shift

of the main peak towards higher binding energies. At the same time, the Zn 2p lines are shifted towards lower BE, which is additional evidence that zinc is built into the framework positions and interacts with the zeolite matrix.

Figure 3 shows the MAS NMR spectra of ZnAlBEA, ZnAlSiBEA, and ZnSiBEA. In all samples, we observe signals around −115 ppm, which correspond to silicon atoms in a $Si(OSi)_4$ environment (named $Q^4$) located in different crystallographic sites [46]. While the resonances are broad for the ZnAlBEA and ZnAlSiBEA samples, for ZnSiBEA, the resonances are narrow. This increase in resolution may be related to the dealumination and, thus, the departure of aluminum atoms, and the incorporation of zinc atoms could also have an effect on the resolution of the different contributions of the $Si(OSi)_4$ species. For ZnAlBEA, the DP MAS NMR spectrum shows a broad signal around −100 ppm composed of two contributions. The first one, at −103.5 ppm, corresponds to $Si(OSi)_3(OAl)$ species, and the second one, at −101.5 ppm, to $Si(OSi)_3(OH)$ species [47,48]. This last contribution is highlighted by the CPMAS experiments since it is strongly exalted in a non-quantitative way. We observe a decrease in this large signal for ZnAlSiBEA and an even larger decrease for ZnSiBEA. This decrease is due to both the departure of aluminum ions and the reaction between the zinc ions and the silanols of the vacant T-atom sites. In the CP spectra, a small fraction of Si atoms in a $Si(OH)_2(OSi)_2$ environment is also highlighted by the peak at 92.0 ppm.

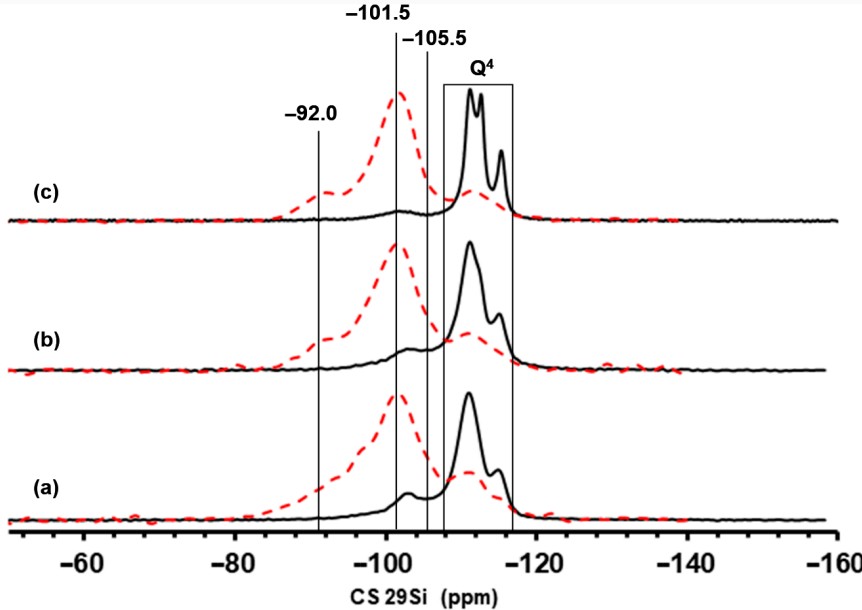

**Figure 3.** $^{29}Si$ MAS NMR spectra of ZnAlBEA (**a**), ZnAlSiBEA (**b**), and ZnSiBEA (**c**). Black curves are direct polarization (DP) spectra and red dotted curves are cross polarization (CP) spectra.

Figure 4 shows the data (profiles) for studying Zn-BEA zeolites through the $NH_3(CO_2)$-TPD technique, and corresponding acid–base characteristics are presented in Table 4.

**Table 4.** Acid–base characteristics of zeolite samples of Zn-BEA according to TPD-$NH_3(CO_2)$ profiles.

| Sample | Concentration of Acidic Sites, rel. un. [1] | | | | Concentration of Basic Sites, rel. un. [1] | | |
|---|---|---|---|---|---|---|---|
| | Weak (293–423 K) [2] | Medium Strength (423–673 K) [2] | Strong (>673 K) [2] | Total | Weak (293–423 K) [2] | Medium Strength (423–673 K) [2] | Total |
| ZnAlBEA | 0.26 | 0.36 | 0.38 | 1.00 | 0.78 | 0.10 | 0.88 |
| ZnAlSiBEA | 0.14 | 0.38 | 0.08 | 0.60 | 0.89 | 0.11 | 1.00 |
| ZnSiBEA | 0.04 | 0.31 | 0.08 | 0.43 | 0.22 | 0.26 | 0.48 |

[1] Rel. un. (Related unit)—ratio of the peak area over a certain temperature range to the peak area under the curve corresponding to the sample with maximum acidity/basicity; [2] desorption temperatures of $NH_3$ and $CO_2$.

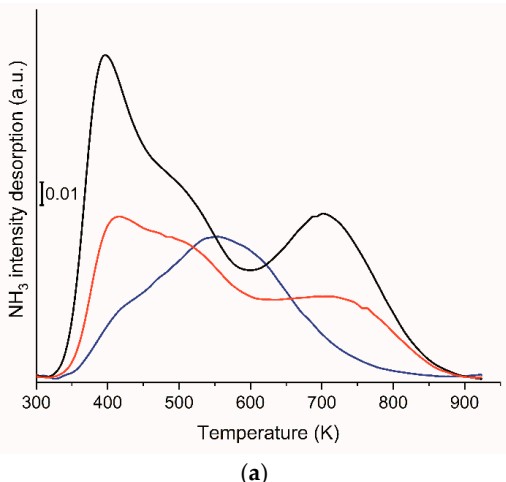
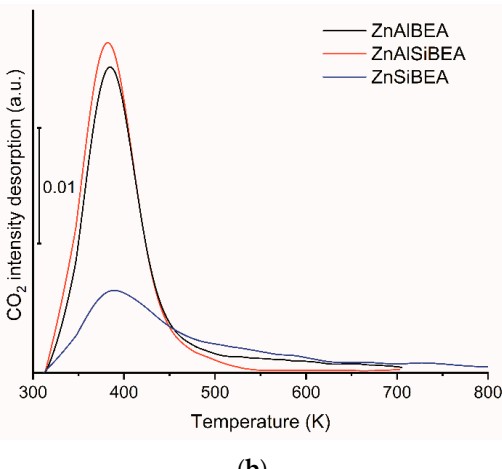

**Figure 4.** Normalized NH3-TPD (**a**) and $CO_2$-TPD (**b**) profiles for the ZnAlBEA, ZnAlSiBEA, and ZnSiBEA compositions.

The ZnAlBEA sample is characterized by the highest total acidity among the zeolites tested. The shape of the TPD-$NH_3$ profile for the ZnAlSiBEA sample is similar to ZnAlBEA, but its total acidity is lower (0.60), which is likely due to a reduction in the number of acid sites formed with the participation of Al(III), which is consistent with [7,49]. In the case of ZnSiBEA, the lowest total acidity is observed (0.43). In addition, the $NH_3$-TPD profile for ZnSiBEA has a significantly different shape compared to that of the samples containing Al. Considering the weak acidity of the SiBEA surface [7,50], the acid sites present on ZnSiBEA may be formed predominantly by Zn(II), incorporated into vacant T-atom sites of the dealuminated BEA zeolite.

Analyzing the $CO_2$-TPD data for the as-prepared Zn-BEA compositions, it should be noted that the corresponding profiles of carbon dioxide desorption for all studied samples have a single maximum in the temperature range of 383–388 K (Figure 4). The surface of both ZnAlBEA and ZnAlSiBEA exhibits predominantly weak basic sites, while the surface of the completely dealuminated ZnSiBEA specimen is characterized by a larger fraction of basic sites of medium strength (Table 4). A more intense shoulder is observed in the $CO_2$-TPD profile for the ZnSiBEA sample at temperatures above 423 K compared to the profiles for ZnAlBEA and ZnAlSiBEA. The function of the medium basic sites of medium strength for the ZnSiBEA specimen is likely performed by oxygen anions/vacancies of [Zn-O-Si] structures at the T-positions of the fully dealuminated zeolite or ZnO particles, dispersed on the SiBEA surface.

The FTIR spectra of pyridine adsorbed on zeolite samples are presented in Figure 5.

The spectra contain the absorption bands (a.b.) of the skeletal vibrations of the heteroaromatic ring (1446 (shoulder), 1453, 1454, 1456, 1490, 1495 (shoulder), 1578, 1600 (shoulder), 1612, 1614, 1616, and 1622 (shoulder) $cm^{-1}$) [14,51]. At the same time, a. b. 1446, 1578, 1600, and 1622 $cm^{-1}$ disappear after evacuation at 523 and 623 K, which allows us to attribute them to weakly bound pyridine, likely through hydrogen bonding with OH groups on the surface of the zeolite samples. The a.b. at 1453–1456, 1490, 1495, and 1612–1616 $cm^{-1}$ refer to pyridine coordinated to Lewis acid sites of the surface [14,15]. Taking into account previous results on the zeolites SiBEA, AlBEA [50,52], AlSiBEA [49], Zn/H-BEA [53], and ZnZr-SiBEA [54], the bands at 1456 $cm^{-1}$ can be attributed to LAS formed with $Al^{3+}$ cations, whereas the bands at 1454–1453 $cm^{-1}$ can be attributed to LAS formed with $Zn^{2+}$. The FTIR-Py spectra of ZnAlBEA and ZnAlSiBEA samples show a. b. at 1547 and 1638 $cm^{-1}$, relating to the pyridinium ion ($PyH^+$) [14,49,51], which indicates the presence of BAS, due to the presence of bridging OH groups bound to aluminum cations at the T-positions of the zeolite framework. The decrease in the intensity of these bands after the heat treatment of these samples in a vacuum at 523 and 623 K is due to the desorption of pyridine associated with BAS of weak/medium strength. The higher intensity of a. b.

at 1453–1456, 1490, and 1616 cm$^{-1}$ in the FTIR spectra of pyridine adsorbed at 423 K on ZnAlBEA and ZnAlSiBEA compared to the corresponding bands for ZnSiBEA is caused by the higher total concentration of acid centers on the surface of the samples based on the initial and partially dealuminated zeolite (Table 4).

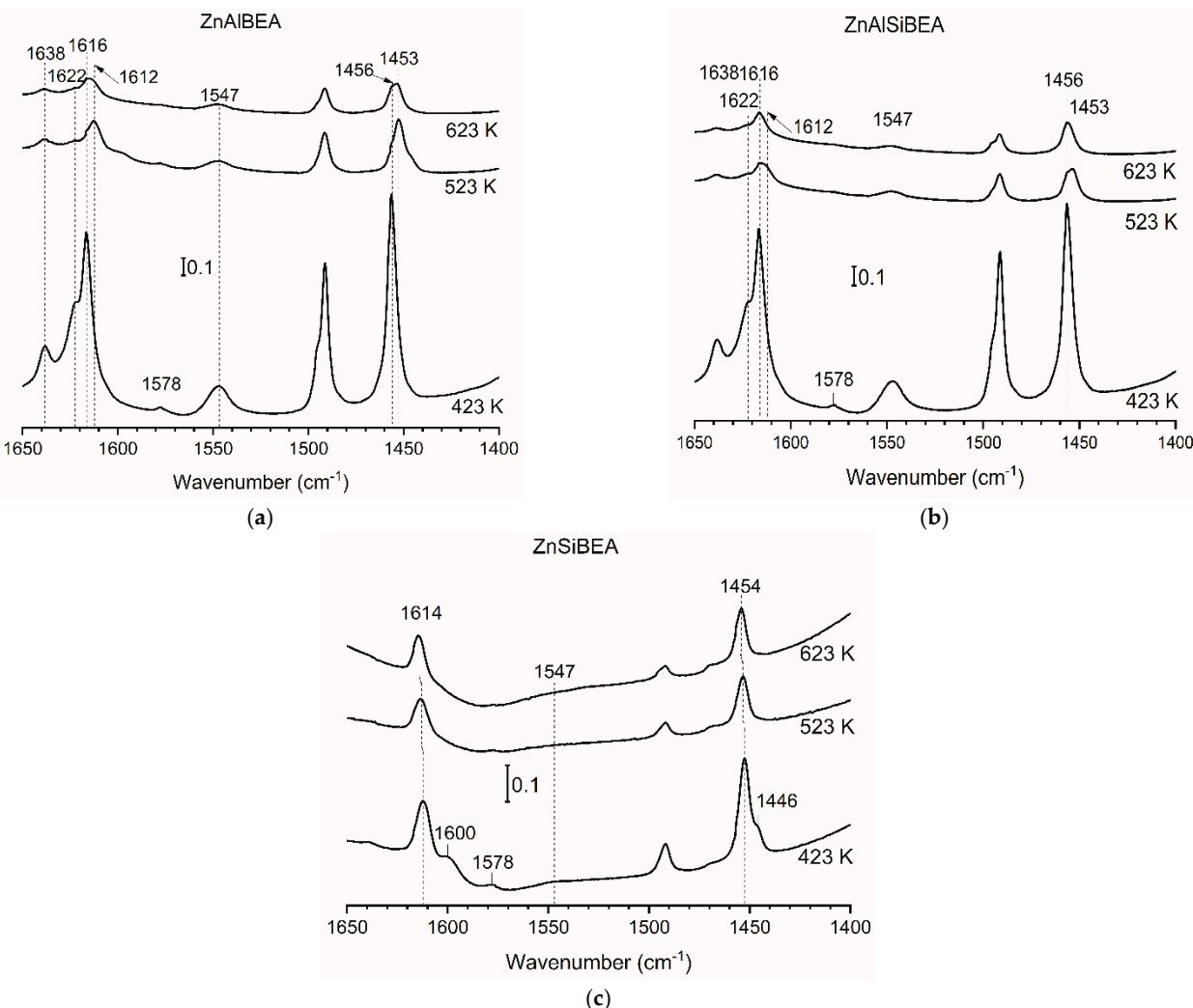

**Figure 5.** Normalized FTIR absorption spectra of pyridine adsorbed at 423 K on ZnAlBEA (**a**), ZnAlSiBEA (**b**), and ZnSiBEA (**c**) after heat treatment at 873 K and subsequent evacuation at different temperatures.

Thus, according to the analysis of FTIR-Py data, the preliminary complete dealumination of BEA zeolite followed by the incorporation of zinc atoms ensures the formation of the ZnSiBEA composition, on the surface of which there are LAS (Zn(II) (likely bound to silanol groups) and BAS are absent, whose function on the surface of ZnAlBEA and ZnAlSiBEA is realized by bridging OH groups bound to aluminum cations at the T-positions of the zeolite framework.

## 2.2. Catalytic Properties of Zn-BEA Zeolites in Propane Dehydrogenation

According to the results obtained, the initial propane conversions for the ZnAlBEA and ZnAlSiBEA catalyst samples exceed those in the presence of the fully dealuminated ZnSiBEA composition (Table 5 and Figure 6). At the same time, for ZnAlSiBEA and ZnAlBEA specimens at 873 and 923 K, propane conversion decreases quite rapidly with increasing TOS, whereas for ZnSiBEA, some decrease in $X_{C3H8}$ with increasing TOS is observed only at 923 K. At temperatures of 873–923 K, the highest formation selectivities

(86–94%) and propene yields (16–18% at 873 K and 30–33% at 923 K) are obtained in the presence of the ZnSiBEA catalyst. The change in selectivity on propene in the ZnSiBEA, ZnAlSiBEA, and ZnAlBEA series occurs symbatically with a change in the Si/Al ratio (1000, 100, and 17, respectively). The highest propene yield is achieved in the presence of the ZnSiBEA catalyst.

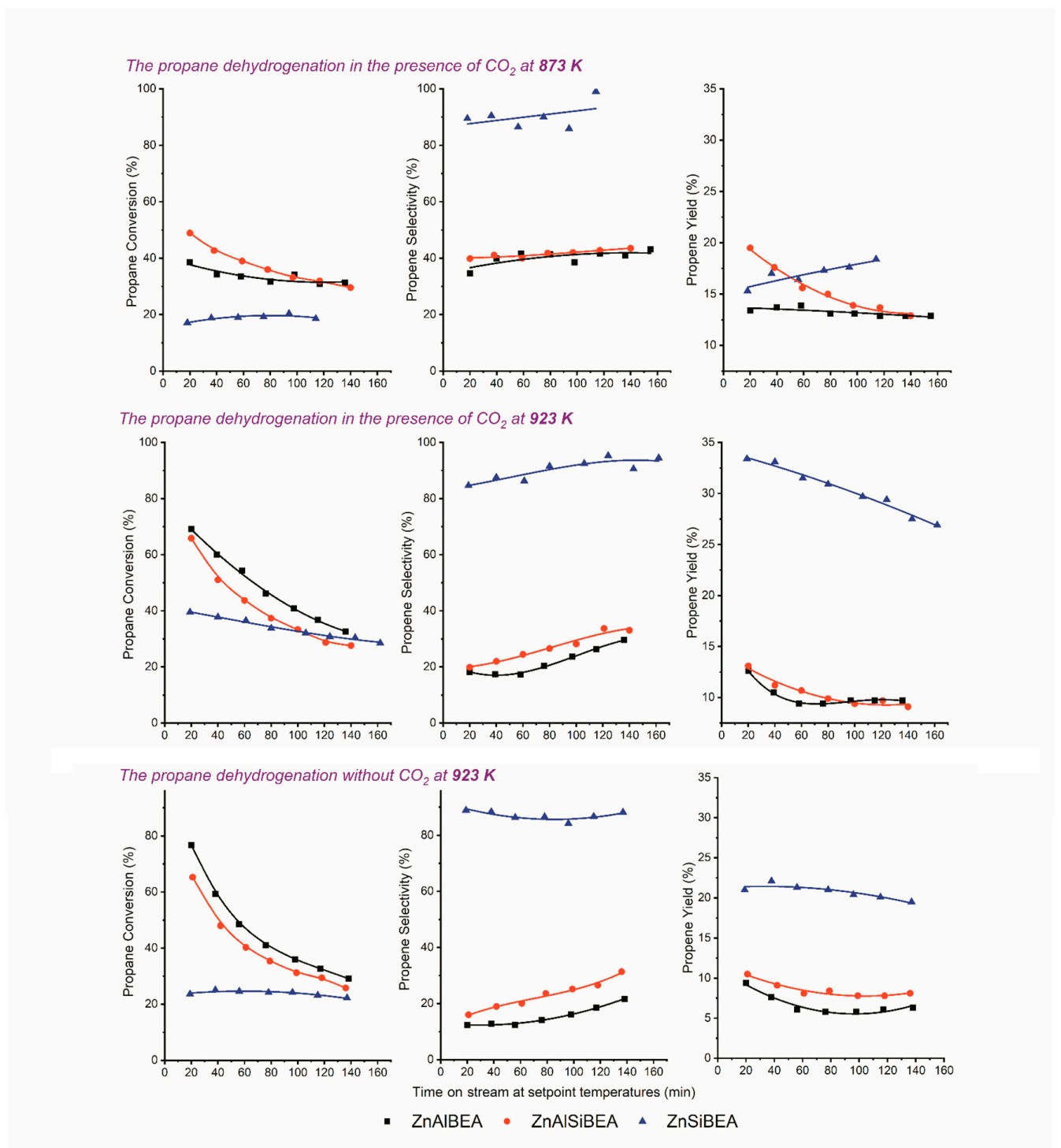

**Figure 6.** Indices of the propane dehydrogenation in the presence/absence of $CO_2$ on the Zn-BEA catalysts.

**Table 5.** Indices of the propane dehydrogenation in the presence of $CO_2$ on Zn-BEA catalysts for different TOS.

| Catalyst | TOS, min | 823 K | | | 873 K | | | 923 K | | |
|---|---|---|---|---|---|---|---|---|---|---|
| | | $X_{C3H8}$ | $S_{C3H6}$ | $Y_{C3H6}$ | $X_{C3H8}$ | $S_{C3H6}$ | $Y_{C3H6}$ | $X_{C3H8}$ | $S_{C3H6}$ | $Y_{C3H6}$ |
| ZnAlBEA | 30 | 38 | 25 | 9.5 | 36 | 37 | 13.3 | 64 | 17 | 10.9 |
| | 60 | 34 | 29 | 9.9 | 33 | 41 | 13.5 | 52 | 18 | 9.4 |
| | 120 | – | – | – | 31 | 42 | 13.0 | 35 | 27 | 9.5 |
| ZnAlSiBEA | 30 | 29 | 36 | 10.4 | 45 | 40 | 18.0 | 58 | 21 | 12.2 |
| | 60 | 23 | 42 | 9.7 | 39 | 41 | 16.0 | 44 | 24 | 10.6 |
| | 120 | – | – | – | 31 | 43 | 13.3 | 29 | 33 | 9.6 |
| ZnSiBEA | 30 | 8 | 57 | 4.6 | 18 | 90 | 16.2 | 38 | 86 | 32.7 |
| | 60 | 9 | 57 | 5.1 | 20 | 90 | 18.0 | 36 | 88 | 31.7 |
| | 120 | – | – | – | 19 | 94 | 17.9 | 32 | 94 | 30.1 |

The by-products of propane conversion are mainly methane and others of propane and propene cracking; coking of the catalyst surface was also observed. The analysis of the obtained data regarding the by-products of the $CO_2$-PDH process indicates that a greater amount of by-products ($CH_4$, $C_2H_6$, $C_2H_4$) is fixed on the Al-containing catalysts. According to the number of by-products formed, the catalysts are arranged in the following order: ZnAlBEA > ZnAlSiBEA > ZnSiBEA.

In the context of the above, it should be noted that Brønsted acid sites, as is well known, intensify the cracking and oligomerization of the olefins including propene, followed by carbonization and, accordingly, in this case, blocking of the active centers of the ZnAlBEA and ZnAlSiBEA samples. Thus, the better catalytic performance of the zeolite composition ZnSiBEA is caused by the absence of BAS on its surface. We note, however, that the absence of Brønsted acid sites does not exclude the possibility of side reactions on the ZnSiBEA catalyst, which are less intense anyway compared to the ZnAlBEA and ZnAlSiBEA samples.

To determine the effect of BEA dealumination on Zn-BEA coking in the $CO_2$-PDH process, $O_2$-TPO profiles of spent catalysts were obtained (120 min at 923 K, at which propane cracking is more intense). The results are shown in Figure 7. Assuming that the intensity of the $O_2$-TPO curves is proportional to the calcined coking products, the area under the $O_2$-TPO curve and the maximum temperature are comparable for the ZnAlSiBEA and ZnAlBEA samples. This is consistent with similar propane conversion/propene yield values at TOS = 120 min (Figure 7). For ZnSiBEA, the area under the $O_2$-TPO curve is much smaller compared to that for ZnAlSiBEA and ZnAlBEA, which is consistent with the greater stability of the ZnSiBEA catalyst compared to Al-containing samples.

The more intense coking of the ZnAlSiBEA and ZnAlBEA samples (as opposed to ZnSiBEA) may be due to the greater number of adsorption centers and their holding capacity. The correlation between the ability to retain propene (according to $C_3H_6$-TPD) and the coking of the catalyst was shown in [24].

To evaluate the number and strength of the propane and propene adsorption centers under reaction conditions, a TPD study of propane and propene (after their adsorption on the catalyst surface from the mixture of propane and propene) was performed.

The results obtained (Figure 8) indicate that the surface of ZnAlBEA has the highest number of centers capable of retaining propane and propene.

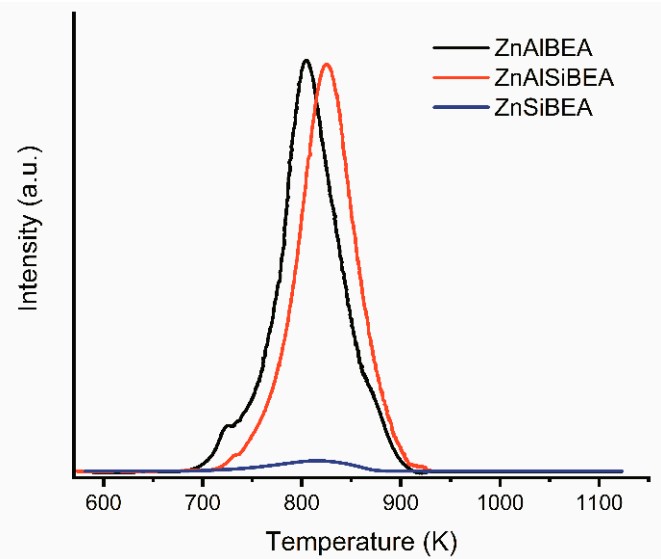

**Figure 7.** Normalized $O_2$-TPO profiles of Zn-BEA catalysts after 2.5 h in the $CO_2$-PDH reaction mixture at 923 K.

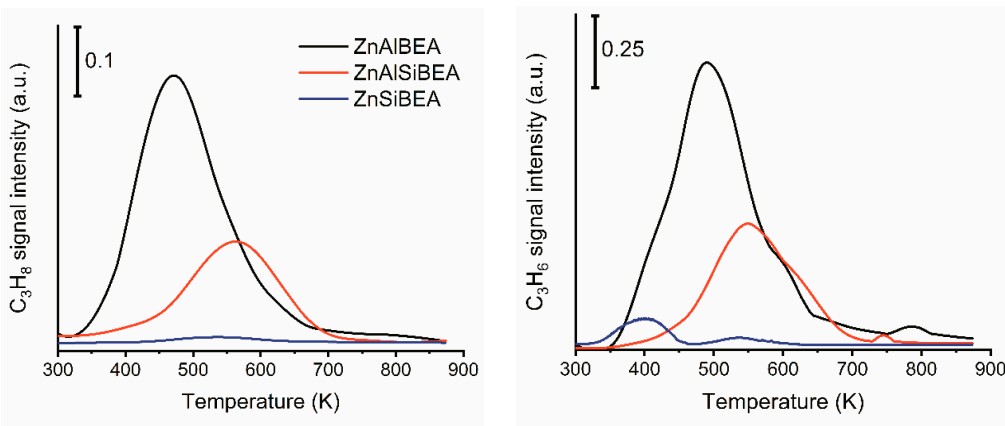

**Figure 8.** Normalized TPD profiles of propane and propene for the Zn-BEA catalysts.

On the ZnAlSiBEA surface, the number of such centers is smaller, although quite significant, especially the strong centers, as indicated by the temperature of the desorption maximum. In the case of ZnSiBEA, the number of such centers is the smallest (compared to ZnAlSiBEA and ZnAlBEA). Therefore, it can be assumed that the surface coking of Zn-BEA samples occurs primarily as a result of propane/propene cracking on Al-containing sites (especially BAS). Brønsted acid sites are capable of protonating (in the case of ZnAlBEA and ZnAlSiBEA) the produced olefin in the π-bond with the formation of carbenium cations $C_3H_7^+$, thus intensifying the course of the side reactions of cracking, oligomerization, and carbonization of the catalyst surface [45,55,56]. The observed result is consistent with the fact that high acidity can be detrimental to the selectivity of olefins due to the difficulty in desorption of the resulting intermediates [5]. The presence of only strong adsorption centers on the ZnAlSiBEA surface (Figure 8) may explain the shift of the maximum on the $O_2$-TPO profile compared to ZnAlBEA (Figure 7).

In general, the initial propane conversion on Zn-BEA catalysts decreases as the Si/Al ratio increases, while propene selectivity and catalyst stability improve. Similar trends of decreasing initial propane conversion and changing selectivity of the $CO_2$-PDH process were found on ZnO/HZSM-5 [14] and $Ga_2O_3$/HZSM-48 [57] catalysts, as well as in the PDH process on ZnO/HZSM-5 [13]. Given the significant amount of BAS on the surface of ZnAlSiBEA and ZnAlBEA, it can be assumed that the formation of propane cracking

products is catalyzed primarily by BAS, which, according to Dzwigaj et al. [23], contribute to the side reactions of oligomerization, alkylation and cracking in the dehydrogenation of alkanes on zeolites.

As noted above, the ZnSiBEA sample is characterized by a more developed mesoporous structure and specific surface area (BET) in comparison with other compositions (Table 1), which contributes to the target $CO_2$-PDH process. However, most important factor determining the catalytic performance of a fully dealuminated specimen is its acid–base characteristics. The ZnSiBEA sample is characterized by the presence of LAS ($Zn^{2+}$) and basic sites ($O^{2-}$ anions and oxygen vacancies) of mostly medium strength on the surface, as well as the absence of Brønsted acid sites, intensifying side reactions with subsequent carbonization of the catalyst surface. The higher concentration of medium-strength basic sites for ZnSiBEA may be a favorable precondition for the most likely route of alkane molecule activation at acid–base paired sites—through dissociation of the $C^{\delta-}–H^{\delta+}$ bond (as a rule, the limiting stage of the propane DH [1–4,58–61]) via deprotonation by nucleophilic $O^{2-}$ anions (Brønsted base sites) and subsequent coordination of the formed carbanions $C_3H_7^-$ with $Zn^{2+}$ cations (LAS), also predominantly of medium strength. In this regard, it should be emphasized that Brønsted base sites play an important role in propane transformations, as they are the ones that ensure the heterolysis of the $C^{\delta-}–H^{\delta+}$ bond and, thus, "release" the electron pair for the coordination of propyl anions with LAS. The basic properties of the most propene-selective ZnSiBEA sample also facilitate carbon dioxide activation, involving the formation of $CO_2^{\cdot-}$ radical anions [62,63] with dissociative adsorption on oxygen vacancies of the zeolite framework lattice [64]. In order to determine the effect of $CO_2$ on $X_{C3H8}$, $S_{C3H6}$, and $Y_{C3H6}$ targets for all catalysts, a propane dehydrogenation reaction was performed in the absence of $CO_2$ in the reaction medium at 923 K (the temperature at which the highest rate of decrease in targets with increasing TOS is observed). The results shown in Figure 6 indicate a positive effect of $CO_2$ on propane conversion, formation selectivity, and propene yield for all Zn-BEA catalyst samples. In the case of ZnAlBEA at 923 K, the presence of $CO_2$ also contributes to a decrease in the catalyst deactivation; thus, the reduction of $X_{C3H8}$ at TOS = 20 min → 140 min in the $CO_2$-PDH reaction is 70 → 32, and in the PDH reaction is 77 → 30.

Thus, the results obtained demonstrate the benefits of producing propene from propane in the presence of carbon dioxide. It is noteworthy that the direct dehydrogenation of propane to propene $C_3H_8 \leftrightarrow C_3H_6 + H_2$ is the reaction with volume increasing, whereas the reverse water–gas shift $CO_2 + H_2 \leftrightarrow CO + H_2O$ is a molecularity-invariant reaction; both are endothermic. As a result, a higher temperature and higher $CO_2$ concentration are thermodynamically favorable for a higher propane conversion to propene and greater assistance of $CO_2$ through the RWGSR. The equilibrium yield of olefin approaches 100% at 973 K and an initial molar ratio of $CO_2/C_3H_8 = 5/1$ according to thermodynamic calculations [65]. However, the selectivity with reference to propene is significantly reduced by sintering, agglomeration, and surface carbonization, thus causing catalyst deactivation. Therefore, the optimal temperature range for propene formation is 873–923 K.

A more detailed characterization of the effect of the Si/Al ratio in Zn-BEA zeolites on the propane conversion and average propene yield in $CO_2$–PDH and PDH reactions over Zn-BEA at TOS = 20 min → 140 min (T = 923 K) is shown in Figure S4. According to the results of $C_3H_8$ dehydrogenation and cracking studies on Zn/H-MFI catalysts [66], their higher activity and selectivity for $C_3H_6$ formation in the absence of cofeed $H_2$ (or $H_2$ removal by $CO_2 + H_2 = CO + H_2O$ reaction) may be a consequence of the conversion of $[ZnH]^+$ cations into bridging $Zn^{2+}$ cations.

Figure 9 and Figure S5 show the temperature dependence of propane conversion and the selectivity of propene formation and yield in the propane dehydrogenation process in the presence/absence of $CO_2$ in the reaction medium for the ZnSiBEA catalyst, which provides better propene yields with stable operation over time.

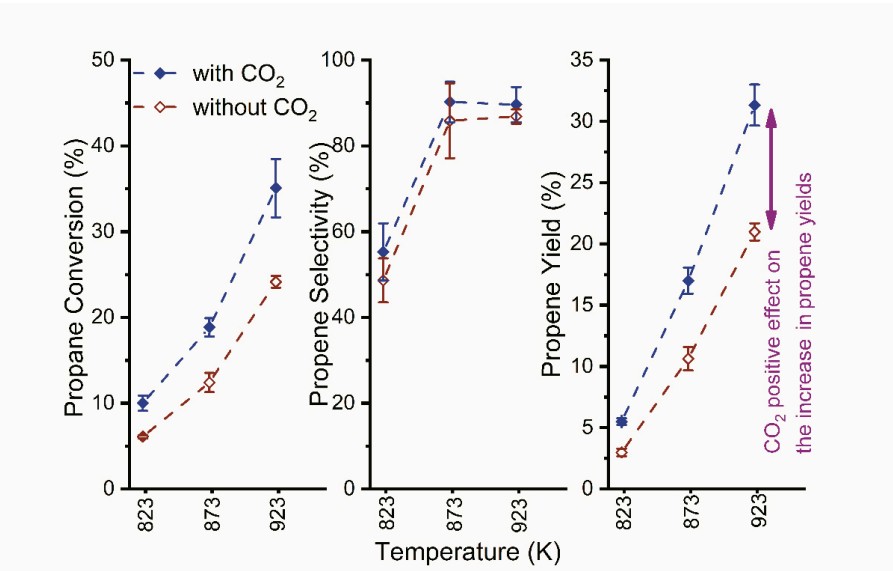

**Figure 9.** Mean values (for TOS = 20–120 min) of propane conversion, propene selectivity, and yield of the propane dehydrogenation in the presence/absence of $CO_2$ on the ZnSiBEA catalyst.

These results clearly indicate a positive effect of $CO_2$ on the production of the target product, propene. At the same time, the presence of $CO_2$ in the reaction medium does not counteract the coking of the catalyst, as evidenced by the slopes of the curves of changes in propane conversion and propene yield over time (dependence of $X_{C3H8}$ and $Y_{C3H6}$ on TOS).

When $CO_2$ is introduced into the reaction mixture, the apparent activation energy of propene formation, determined from Arrhenius plots (Figure S6), decreases from $124 \pm 9$ kJ·mol$^{-1}$ to $110 \pm 7$ kJ·mol$^{-1}$ (it is worth noting that the calculated $E_a$ values are close to those obtained in [67], but for $ZrO_2$-based catalysts).

## 3. Materials and Methods

### 3.1. Zeolite Sample Preparation and Characterization

Zeolite samples were prepared based on templated tetraethylammonium BEA zeolite (TEABEA, Si/Al = 17) manufactured by Research Institute of Petroleum Processing (RIPP), Haidian District, Beijing, China.

To obtain the exemplary starting AlBEA zeolite, the TEABEA was calcined in air at 550 °C for 15 h. To obtain partially and fully dealuminated zeolite, the starting AlBEA zeolite was treated with $HNO_3$ solution (6 or 13 mol·L$^{-1}$) for 4 h at 353 K according to the method described in [49,68]. Partially and fully dealuminated zeolites with Si/Al = 100 (AlSiBEA) and 1000 (SiBEA) compositions were separated through centrifugation, washed with distilled water, and dried for 24 h at 353 K.

In order to introduce 1 wt % of Zn into the AlBEA, AlSiBEA, and SiBEA zeolites, these samples were treated with excess aqueous zinc(II) nitrate solutions at pH 3.0 to obtain the ZnAlBEA, ZnAlSiBEA, and ZnSiBEA series.

X-ray diffraction patterns of the prepared zeolite specimens were recorded on a Bruker AXS GmbH D8 Advance diffractometer (series II) (nickel filter, CuK$_\alpha$ radiation, $\lambda$ = 0.154184 nm).

The texture characteristics ($S_{BET}$, pore volume and size, mesopore surface) of the studied samples were determined using $N_2$ ad/desorption at a low temperature (77 K) on a Sorptomatic 1990 porous materials analyzer with preliminary evacuation (573 K, 0.001 Pa/7.50 Torr). The pore size distribution was calculated using the Saito–Foley (micropores) and Barrett–Joyner–Halenda (mesopores) methods, and the volume of micropores and specific surface of mesopores were determined using the t-plot method.

The morphology of the samples was determined out using a JEOL JSM–7500F Field Emission Scanning Electron Microscope (JEOL, Akishima, Japan) equipped with a re-

tractable backscattered-electron detector (RBEI) and energy-dispersive spectra (EDS) detection system of a characteristic X-ray radiation Ztec Live for EDS system (Oxford Instruments, Abingdon, London, UK).

The X-ray photoelectron spectroscopy (XPS) investigations were carried out in a multi-chamber ultra-high-vacuum system equipped with a hemispherical analyzer (SES R4000, Gammadata Scienta). A Mg $K_\alpha$ X-ray source (1253.6 eV) was used for photoelectron generation. The anode was operated at 180 W (12 kV, 15 mA). The spectrometer was calibrated according to ISO 15472:2001. The energy resolution of the system (pass energy 100 eV) determined for the Ag $3d_{5/2}$ excitation line was 0.9 eV. The base pressure in the analytical chamber was $1 \times 10^{-10}$ mbar and approximately $6 \times 10^{-9}$ mbar during the experiment. The powder samples were examined after being pressed in indium foil and mounted on a special holder. The analysis area of the samples was about 4 mm$^2$ ($5 \times 0.8$ mm). High-resolution spectra were collected at a pass energy of 100 eV (with a 25 meV step), while survey scans were collected at a pass energy of 200 eV (with a 0.25 eV step). The experimental curves were fitted in CasaXPS 2.3.23 using a combination of Gaussian and Lorentzian lines with variable ratios (70:30) after subtracting a Shirley-type background. The relative ratio of the intensities of the $2p_{3/2}$ and $2p_{1/2}$ lines in the doublets was set to 2:1. All binding energies were charge-corrected to the carbon C 1s excitation, which was set to 285.0 eV.

Solid-state magic angle spinning nuclear magnetic resonance (MAS NMR) experiments were performed on a Bruker (Billerica, MA, USA) AVANCE500 spectrometer at 11.7 T in 4 mm zirconia rotors spinning at 14 kHz. $^{29}$Si direct polarization (DP) MAS NMR and $^1$H-$^{29}$Si cross polarization (CP) MAS NMR were performed with a 5 mm zirconia rotor with a 5 kHz spinning speed, 2 μs excitation pulse, and 10 s recycle delay. 3-(trimethylsilyl)-1-propanosulfonic sodium salt was used for setting the Hartmann–Hahn conditions. The proton $\pi/2$ pulse duration, the contact time, and recycle delay were 3 μs, 5 ms, and 5 s, respectively.

One-pass temperature-programmed desorption of propane/propene, carbon dioxide, and/or ammonia ($C_3H_8/C_3H_6$ ($CO_2$, $NH_3$)-TPD) was carried out in an ultra-high-vacuum (UHV) black chamber-type system, controlling the desorbed molecules using a time-of-flight MSX-3PC mass spectrometer (Electron, Iviv, Ukraine). Zeolite samples of 0.02 g each were preheated at 873 K for 2 h under a pressure of $10^{-9}$ Torr and then cooled to room temperature in a vacuum. The adsorption of propane and propene (gas mixture 10% $C_3H_8$ + 10% $C_3H_6$ in He), ammonia (99.99%), and carbon dioxide (99.99%) were carried out with the respective molecular probe gases for 12 h. The programmed temperature rise was carried out at a rate of 9 K·min$^{-1}$. The application of the TPD technique is described in detail in [69].

The acidity and basicity of the samples were evaluated based on signal intensity proportional to the amount of $NH_3$ and $CO_2$ adsorbed on the sample surface at a given temperature, normalized to the sample mass. The areas under the curves for the TPD profiles, corresponding to acidic or basic sites of a given strength on the surface of the samples, were calculated after the deconvolution of the spectrum into a minimum number of components (using a Gaussian distribution) so that the total curve of the deconvoluted spectrum coincided with the experimental curve.

The nature of acid sites on the surfaces of the samples was investigated through Fourier-transform infrared spectroscopy of adsorbed pyridine as a probing molecule (FTIR-Py) using a Spectrum One FTIR spectrometer (Perkin Elmer, Waltham, MA, USA). Samples in the form of thin wafers, pressed from fine powders with suitable catalytic compositions, were pre-heated at 693 K for 1 h under a $10^{-3}$ Torr vacuum in a quartz cuvette reactor. Adsorption of gaseous pyridine was carried out at 423 K and then evacuated at 423, 523, and 623 K for 0.5 h. FTIR spectra of adsorbed Py were recorded at room temperature (spectrometer beam temperature) with a resolution of 1 cm$^{-1}$ and 24 scans.

The temperature-programmed oxidation of the catalysts with $O_2$ ($O_2$-TPO) was performed on an AMI-300Lite Catalyst Characterization Instrument (Altamira Instruments,

Pittsburgh, PA, USA). Prior to testing, samples (0.1 g after 2.5 h of work at 923 K in $CO_2$-PDH) were treated at 573 K for 20 min in He gas at a flow rate of 25 mL·min$^{-1}$. After cooling to 323 K, the samples were immediately reheated in 20 vol% $O_2$ in He from 323 K to 1073 K (a flow rate of 25 mL·min$^{-1}$) with a linear temperature ramp of 5 K·min$^{-1}$. The signal changes of the effluent gases were analyzed using a thermal conductivity detector.

*3.2. Catalytic Activity Measurements*

Catalytic experiments were carried out in a flow-type quartz reactor at atmospheric pressure, at a temperature of 823–923 K, with a gas hour space velocity (GHSV) of 6000 h$^{-1}$ (catalyst loading 0.2 g, grain size of 0.25–0.5 mm, and reaction mixture (RM) flow rate of 30 cm$^3$·min$^{-1}$). The RM composition for $CO_2$-PDH was 2.5 vol. % of $C_3H_8$ and 15 vol. % $CO_2$ in He; for PDH, it was –2.5 vol. % of $C_3H_8$ in He. The weight hour space velocity (WHSV) was 0.4 $g_{C3H8}$·$g_{cat}^{-1}$·h$^{-1}$. The reagents and reaction products ($C_3H_8$, $CO_2$, $C_3H_6$, $CH_4$, $C_2H_4$, $C_2H_6$) were analyzed using gas chromatography (Krystallux 4000M, MetaChrom, Yoshkar-Ola, Russian Federation) equipped with a thermal conductivity detector and a column packed with Porapak Q. The gas sample was preliminarily dried by passing it through a calcium chloride trap.

Before evaluation, the catalysts were pretreated in He flow at the required temperatures for 30 min. It should be noted that the study of catalytic properties was performed with a gradual temperature rise in the range of 823–923 K with a step of 50 K, and with interstage regeneration of the catalyst sample in situ before the RM was introduced into the reactor. Regeneration of the sample was carried out through calcination in air at 873 K for 2 h to remove coke particles.

The catalytic properties of samples in the $CO_2$-PDH and PDH processes were characterized on the basis of propane conversion ($X_{C3H8}$), selectivity ($S_{C3H6}$), and yield ($Y_{C3H6}$) with respect to propene. The indices of the catalytic process were calculated using the following formulas:

$$X_{C3H8} = (C_{C3H8\ inlet} - C_{C3H8\ outlet})/C_{C3H8\ inlet} \cdot 100\%,$$

$$S_{C3H6} = C_{C3H6}/(C_{C3H8\ inlet} - C_{C3H8\ outlet})\ 100\%,$$

$$Y_{C3H6} = X_{C3H8} \cdot S_{C3H6}/100\%,$$

where $C_{C3H8\ inlet(outlet)}$ is the mole concentration of propane at the inlet (outlet) of the reactor and $C_{C3H6}$ is the mole concentration of produced propene.

The propene formation rate was calculated per unit mass of catalyst ($mol_{C3H6}$·$kg_{cat}^{-1}$·s$^{-1}$) as follows:

$$r_{C3H6} = F_{C3H8} \cdot (Y_{C3H6}/100\%)/m_{cat},$$

where $F_{C3H8}$ is the molar flow rate of propane (mol/s) and $m_{cat}$ is the mass of catalyst (kg).

Based on these calculations, the Arrhenius plots for propene formation in the PDH and $CO_2$-PDH processes were drawn, and the associated activation energies were determined from the slopes of the corresponding plots.

## 4. Conclusions

The effect of the preliminary dealumination of BEA zeolite on the acid–base characteristics and catalytic performance of the 1%Zn-BEA compositions in the dehydrogenation of propane in the presence/absence of $CO_2$ was determined.

The post-synthesis procedure of preparing Zn-BEA catalyst samples including partial and full preliminary dealumination of the TEABEA initial specimen (Si/Al = 17) followed by introducing $Zn^{2+}$ cations into vacant T-atom sites of the zeolite framework leads to a reduction in the total acidity of ZnAlSiBEA (0.60, rel. un.) and ZnSiBEA (0.43) compared to ZnAlBEA (1.0).

Full dealumination also results in the development of mesoporous structure and specific surface area (BET) while halving the total basicity of Zn-BEA, creating acid sites (Lewis) and basic sites of predominantly medium strength, and removing Brønsted acid sites from the surface.

In the presence of the ZnSiBEA sample, which has the lowest total acidity and basicity, the achieved selectivity of 86–94% and yield of 30–33% related to propene in the $CO_2$-PDH process (at 923 K) exceed those for the ZnAlBEA and ZnAlSiBEA compositions.

The positive effect of full dealumination on selectivity and lower deactivation with increasing TOS are mainly attributed to the lack of BAS on the surface of ZnSiBEA, which are capable of protonating (in the case of ZnAlBEA and ZnAlSiBEA) the produced olefin in the π-bond and, thus, intensifying the course of the side reactions of oligomerization and cracking with subsequent carbonization of the catalyst surface.

A comparison of the achieved selectivity and yields for propene in the dehydrogenation of propane with/without carbon dioxide demonstrates the advantages of target olefin production in the presence of $CO_2$ using Zn-BEA zeolite catalysts.

**Supplementary Materials:** The following supporting information can be downloaded at: https://www.mdpi.com/article/10.3390/catal13040681/s1, Table S1: XPS data obtained from C 1s and O 1s regions of Zn-BEA zeolites. SEM images with magnification of 25000 (left) and 50000 (right) of (a,b) ZnSiBEA, (c,d) ZnAlSiBEA, and (e,f) ZnAlBEA; Figure S2: XPS spectra of Zn-BEA zeolites in the BE region of Si2p; Figure S3: XPS spectra of Zn-BEA zeolites in the BE region of Al 2p; Figure S4: Impact of Si/Al in Zn-BEA zeolites on the propane conversion and the average propene yield in $CO_2$-PDH and PDH reactions on Zn-BEA (TOS = 20 min → 140 min, T = 923 K). Figure S5: Propane conversion and propene yield versus TOS in the propane dehydrogenation with (full symbols) and without (empty symbols) $CO_2$ on the ZnSiBEA catalyst. Figure S6: Arrhenius plots of propene formation in PDH and $CO_2$-PDH processes on the ZnSiBEA catalyst.

**Author Contributions:** Conceptualization: S.O. and S.D.; Methodology: S.D., P.K., M.Z. and A.K.; Software: V.C.; Validation: V.C.; Formal Analysis: S.O., S.D. and A.K.; Investigation: P.K., V.C., D.B., J.G., Y.M. and S.D.; Resources: S.O. and S.D.; Data Curation: S.O. and S.D.; Writing—Original Draft Preparation: A.K. and P.K.; Writing—Review and Editing: S.O. and S.D.; Visualization: P.K., V.C., D.B., J.G., Y.M. and A.K.; Supervision: S.O. and S.D. All authors have read and agreed to the published version of the manuscript.

**Funding:** This research received no external funding.

**Data Availability Statement:** Suggested Data Availability Statements are available in section "MDPI Research Data Policies" at https://www.mdpi.com/ethics (accessed on 26 March 2023).

**Conflicts of Interest:** The authors declare no conflict of interest.

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
