# Peer review of "CO2-Assisted Dehydrogenation of Propane to Propene over Zn-BEA Zeolites: Impact of Acid–Base Characteristics on Catalytic Performance"

_catalysts, doi:10.3390/catal13040681_

Round 1

Reviewer 1 Report

The study carried out by Stanislaw Dzwigaj with co-authors is applied in nature and reveals the influence of preliminary dealumination of BEA zeolite on the acid-base characteristics of synthesized Zn-BEA samples and their catalytic properties in the CO2-mediated dehydrogenation of propane to propene. Due to its possible practical application in the future, I think that the article may be published in the Catalysts journal. The results obtained are well described and clearly presented. A detailed analysis of the literature was carried out - 69 references. I think the research done is worthy of the Catalysts journal.

Author Response

We thank Reviewer 1 who approved the article.

Reviewer 2 Report

In this work, zinc loaded on dealuminated BEA zeolite was used as a catalyst in the dehydrogenation of propane to propene; the effect of acidity and CO2 assistance on the propane dehydrogenation efficiency was investigated. The results illustrated that the dealumination helps to develop mesoporous structure, remove Brønsted acid sites, and strengthen the Lewis acid sites and basic sites of medium strength, which selectively benefits the dehydrogenation of propane to propene and suppresses the carbonaceous deposition and related side reactions; in addition, the presence of CO2 can further enhance the propane dehydrogenation efficiency.

Such results were interesting and the manuscript is informative. However, it also seems to this reviewer that current manuscript is somewhat too lengthy and there are also some uncertainties. As a result, this reviewer suggests that it may be accepted for publication in the journal of Catalysts after proper revision.

(1) Current manuscript is somewhat too lengthy; the authors may condense the results and discussion and make it more concise. The single sentence paragraphs like “Figure 1 presents X-ray diffraction patterns of Zn-BEA samples” and “Table 1 shows the texture characteristics of the Zn-BEA samples” may be incorporated into subsequent paragraphs and displaced with “as shown in Figure 1” or “as given in Table 1”. 

(2) The basic properties (like XRD patterns, acidity, etc.) of pristine BEA zeolite (AlBEA) and dealuminated ones (AlSiBEA and SiBEA) without loading Zn may be provided and compared with ZnAlBEA, ZnAlSiBEA and ZnSiBEA.

(3) The bulk Si/Al ratios (may be measured by ICP) should be provided.

(4) The dehydrogenation of propane to propene is a volume-increasing reaction, whereas the reverse water-gas shift is molecularity-invariant during reaction; both are endothermic. As a result, a higher temperature and higher CO2 concentration is thermodynamically favorable for a higher propane conversion to propene and greater assistance of CO2 through the reverse water-gas shift. These points should be considered and strengthened in giving the results and discussion.

(5) The use of the words of both “propene” and “propylene” may be avoided.

(6) The state of Zn species in the presence of CO2 may be different from that in the absence of CO2. In addition, the reaction mechanism of propane dehydrogenation concerning CO2 assistance may be considered, by referring the papers like “Fuel, 2013, 109C: 43–48”; “Appl. Catal. A: General, 2006, 302(2): 185–192”; “J. Mol. Catal. A: Chem., 2004, 210(1-2): 189–195”.

(7) A carful proofreading is essential. Line 39, Table 3 mentioned in the text is absent; Lines 166–173, Table 4 does not correspond to the one in current manuscript; Line 123, “as was mentioned above” be “as mentioned in the experimental section”; Al 2P and Si 2P XPS spectra are mentioned but cannot be seen in the manuscript; Line 295, where is the Table 24; etc.

Author Response

Responses to the comments and suggestions of Reviewer 2:

In this work, zinc loaded on dealuminated BEA zeolite was used as a catalyst in the dehydrogenation of propane to propene; the effect of acidity and CO2 assistance on the propane dehydrogenation efficiency was investigated. The results illustrated that the dealumination helps to develop mesoporous structure, remove Brønsted acid sites, and strengthen the Lewis acid sites and basic sites of medium strength, which selectively benefits the dehydrogenation of propane to propene and suppresses the carbonaceous deposition and related side reactions; in addition, the presence of CO2 can further enhance the propane dehydrogenation efficiency.

Such results were interesting and the manuscript is informative. However, it also seems to this reviewer that current manuscript is somewhat too lengthy and there are also some uncertainties. As a result, this reviewer suggests that it may be accepted for publication in the journal of Catalysts after proper revision.

Answer:

Thank you very much for your comments and suggestions, we appreciate them. Please find our answers below whereas the changes are marked in red in the revised manuscript.

(1) Current manuscript is somewhat too lengthy; the authors may condense the results and discussion and make it more concise. The single sentence paragraphs like “Figure 1 presents X-ray diffraction patterns of Zn-BEA samples” and “Table 1 shows the texture characteristics of the Zn-BEA samples” may be incorporated into subsequent paragraphs and displaced with “as shown in Figure 1” or “as given in Table 1”. 

Answer:

In the updated manuscript, the aforementioned single-sentence paragraphs are incorporated into subsequent paragraphs. At the same time, taking into account the requirements of the Catalysts journal for the length of the article (at least 4,000 words), we believe it is inappropriate to reduce the manuscript.

(2) The basic properties (like XRD patterns, acidity, etc.) of pristine BEA zeolite (AlBEA) and dealuminated ones (AlSiBEA and SiBEA) without loading Zn may be provided and compared with ZnAlBEA, ZnAlSiBEA and ZnSiBEA.                                  

Answer:

The basic physicochemical properties of AlBEA and SiBEA (XRD patterns, FTIR, acidity, etc.) are presented in earlier works (refs. [7,16,49,50]) on which we refer, in particular when discussing the acidity of ZnBEA samples.

(3) The bulk Si/Al ratios (may be measured by ICP) should be provided.

Answer:

These Si/Al ratios are present in # 3.1 of the manuscript: AlBEA (17), AlSiBEA (100), and SiBEA (1000). These ratios have been confirmed using ICP method.

(4) The dehydrogenation of propane to propene is a volume-increasing reaction, whereas the reverse water-gas shift is molecularity-invariant during reaction; both are endothermic. As a result, a higher temperature and higher CO2 concentration is thermodynamically favorable for a higher propane conversion to propene and greater assistance of CO2 through the reverse water-gas shift. These points should be considered and strengthened in giving the results and discussion.

Answer:

In the renewed manuscript, the role of the temperature factor and CO2 concentration in the context of thermodynamics on propene production in the CO2-PDH is considered and emphasized when discussing the results.

(5) The use of the words of both “propene” and “propylene” may be avoided.

Answer:

In the updated manuscript only "propene" appears. At the same time, "polypropylene" and "propylene glycol" were left unchanged in the Introduction, since "polypropene" and "propene glycol" are practically not used in the literature.

(6) The state of Zn species in the presence of CO2 may be different from that in the absence of CO2. In addition, the reaction mechanism of propane dehydrogenation concerning CO2 assistance may be considered, by referring the papers like “Fuel, 2013, 109C: 43–48”; “Appl. Catal. A: General, 2006, 302(2): 185–192”; “J. Mol. Catal. A: Chem., 2004, 210(1-2): 189–195”.

Answer:

It should be mentioned that regardless of the presence/absence of CO2 in the reaction mixture, Zn2+ species cannot undergo redox transformations under reaction conditions.

In the renewed article, the reaction mechanism of propane dehydrogenation concerning carbon dioxide assistance is considered. It should be noted in this context that the formation of alkene during CO2-PDH on catalysts, which do not undergo redox transformations under reaction conditions (including Zn-based), proceeds via direct dehydrogenation C3H↔ C3H+ H2, the equilibrium of which shifts in the direction of propene production due to the consumption of hydrogen in the reverse water-gas shift reaction CO+ H↔ CO + H2O. Carbon dioxide may also participate in coke gasification through the reverse Boudouard reaction CO2 + C ↔ 2CO that enhances the catalyst stability.

It should be noted that “the papers like “Fuel, 2013, 109C: 43–48”; “Appl. Catal. A: General, 2006, 302(2): 185–192”; “J. Mol. Catal. A: Chem., 2004, 210(1-2): 189–195” are devoted to other processes and catalysts (in particular, the dehydrogenation of ethylbenzene over supported vanadium catalysts):

S.W. Chen, Z.F. Qin*, G.F. Wang, M. Dong, J.G. Wang*, Promoting effect of carbon dioxide on the dehydrogenation of ethylbenzene over silica-supported vanadium catalysts, Fuel, 2013, 109C: 43–48.

Chen, Zhangfeng Qin, Xiufeng Xu, Jianguo Wang* Structure and properties of the alumina-supported vanadia catalysts for ethylbenzene dehydrogenation in the presence of carbon dioxide, Appl. Catal. A: General, 2006, 302(2): 185–192.

Ailing Sun, Zhangfeng Qin, Shuwei Chen, Jianguo Wang. Role of carbon dioxide in the ethylbenzene dehydrogenation coupled with reverse water–gas shift, J. Mol. Catal. A: Chem., 2004, 210(1-2): 189–195.

(7) A careful proofreading is essential. Line 39, Table 3 mentioned in the text is absent; Lines 166–173, Table 4 does not correspond to the one in current manuscript; Line 123, “as was mentioned above” be “as mentioned in the experimental section”; Al 2P and Si 2P XPS spectra are mentioned but cannot be seen in the manuscript; Line 295, where is the Table 24; etc.

Answer:

We are sorry for these mistakes.

We have taken all these comments and suggestions into account including returning Table 3 to the article, which presents the characteristics of Al 2P and Si 2P XPS spectra. The Al 2P and Si 2P XPS spectra are presented in Supplementary Materials

Reviewer 3 Report

Recommendation: Publish after major revisions noted

Comments:
In this manuscript, Orlyk and co-workers presented the effect of preliminary dealumination of BEA zeolite on the acid-base characteristics and catalytic performance of the 1%Zn-BEA compositions in the dehydrogenation of propane under the presence/absence of CO2. The authors clearly showed that the full dealumination of the catalyst could reduce both acidity and basicity, which appeared to improve both selectivity and stability. However, many of their catalysts did not achieve the steady state in the performance tests, which led to inaccurate comparisons with respect to selectivity and activity. In addition, more characterizations need to be performed to understand what is going on with Zn in the catalysts. While the overall conclusions are reasonable, the manuscript has several deficiencies that need to be addressed before publication.

Detailed comments are listed below:

1)    No information about the elemental distribution of Zn in each catalyst.

2)    What is the particle size of each catalyst? What is the status of Zn?

3)    In Figure 6, the reaction time is too short. Only ZnSiBEA achieved its steady state pretty quickly in the 1st and 3rd reaction condition. The rests were not at the steady state. This means the catalysts were still changing under those reaction conditions, which also made Figures 9 and S2 be inaccurate. The authors need to extend their reaction duration to show each catalyst reaches the steady state before making any conclusions from catalytic performance.

Author Response

Responses to the Comments and Suggestions of Reviewer 3:

In this manuscript, Orlyk and co-workers presented the effect of preliminary dealumination of BEA zeolite on the acid-base characteristics and catalytic performance of the 1%Zn-BEA compositions in the dehydrogenation of propane under the presence/absence of CO2. The authors clearly showed that the full dealumination of the catalyst could reduce both acidity and basicity, which appeared to improve both selectivity and stability. However, many of their catalysts did not achieve the steady state in the performance tests, which led to inaccurate comparisons with respect to selectivity and activity. In addition, more characterizations need to be performed to understand what is going on with Zn in the catalysts. While the overall conclusions are reasonable, the manuscript has several deficiencies that need to be addressed before publication.

Answer:

Thank you very much for your comments and suggestions, we appreciate them. Please find our answers below whereas the changes are marked in red in the revised manuscript.

(1) No information about the elemental distribution of Zn in each catalyst.

Answer:

The surface elemental distribution of Zn in each catalyst (obtained from the XPS data) is shown in Table 2.

(2) What is the particle size of each catalyst? What is the status of Zn?

Answer:

Zn2+ cations incorporated into the T positions of the zeolite framework as Zn(II) species are a catalytically active sites. At the same time, its small content (1 % wt) within the Zn-BEA samples makes it impossible for the corresponding reflections to appear in the XRD patterns, as well as the calculation of the average size of corresponding particles using the Scherrer equation.

(3) In Figure 6, the reaction time is too short. Only ZnSiBEA achieved its steady state pretty quickly in the 1st and 3rd reaction condition. The rests were not at the steady state. This means the catalysts were still changing under those reaction conditions, which also made Figures 9 and S3 be inaccurate. The authors need to extend their reaction duration to show each catalyst reaches the steady state before making any conclusions from catalytic performance.

Answer:

In our opinion, it is not worth talking about (not)reaching a steady state (quasi-stationary state) of catalysts in the CO2-PDH/PDH process during their stay in the reaction stream (up to 2.5 hours). The data of Table 5 and Fig. 6 testify to a different influence of the reaction medium on the activity of the catalysts. A number of reviews and original papers on this process usually give initial values (alkane conversion, selectivity, and olefin yield) and/or average values achieved over time on stream (TOS).

The results for each of the ZnAlBEA, ZnAlSiBEA, and ZnSiBEA samples are obtained under the same conditions of the catalytic runs (reaction mixture, temperature range, catalyst samples loading, GHSV, and WHSV) which gives grounds for correct comparison of the catalytic behavior. And these results clearly testify to the better catalytic performance of a fully dealuminated specimen ZnSiBEA: regardless of the reaction duration, the selectivity and yield with respect to propene for the ZnAlBEA and ZnAlSiBEA samples will be inferior to those for ZnSiBEA.

It should also be noted that the data presented in Figures 9 and S4 refer to only the ZnSiBEA catalyst.

Round 2

Reviewer 2 Report

It seems that most of the issues raised by the reviewers to the old version manuscript have been properly responded in the revised manuscript. As a result, this reviewer suggests that the revised manuscript may now be acceptable for publication in the journal of Catalysts.

Author Response

Thank you very much for acceptation our revised manuscript for publication in Catalysts journal

Reviewer 3 Report

Recommendation: Reject

Comments:
The authors are not able to address most comments noted from the last review cycle. The current results don’t provide the information of Zn distribution and particle size. The necessary EDX and TEM measurements are missing. I totally disagree with the authors’ response to my last comment. Comparing catalytic performance over different catalysts at the steady state (or near the steady state at least) is very critical in heterogeneous catalysis. Without a clear improvement of this manuscript, I think it is not suitable for this journal.

Author Response

In Figure S1 the secondary electron images of the three samples ZnSiBEA, ZnAlSiBEA and ZnAlBEA are presented. They illustrate the morphology of the examined samples. SEM/EDS analysis reveals that they are composed of Si, O,  Zn, and Al and the amount of Al increases from 0.4 wt % in ZnSiBEA to 1.2 wt % in  ZnAlSiBEA and 2.6 wt % in ZnAlBEA.  The amount of Zn is close to 1.2 wt %, however, ZnSiBEA exhibits the highest quantity of zinc  (1.4 wt %). It is worth noticing that the ZnSiBEA sample exhibits the presence of the biggest 160-100 nm particles, while ZnAlSiBEA composed the crystallites not exceeding 120 nm. The morphlogy of ZnAlBEA particles is very similar and in the range of 150-100 nm.